# Prevalence of Falls on Mount Fuji and Associated with Risk Factors: A Questionnaire Survey Study

**DOI:** 10.3390/ijerph16214234

**Published:** 2019-10-31

**Authors:** Tadashi Uno, Masaya Fujino, Atsushi Ohwaki, Masahiro Horiuchi

**Affiliations:** 1Division of Human Environmental Science, Mt. Fuji Research Institute, Kami-Yoshida 5597-1, Fuji-Yoshida-City, Yamanashi 403-0005, Japan; unochu@mfri.pref.yamanashi.jp (T.U.); masayafujino@mfri.pref.yamanashi.jp (M.F.); 2Division of Natural Environmental Science, Mt. Fuji Research Institute, Kami-Yoshida 5597-1, Fuji-Yoshida-City, Yamanashi 403-0005, Japan; ohwakiat@mfri.pref.yamanashi.jp

**Keywords:** climbing experience, high altitude, multiple logistic regression, sex difference, subjective feelings

## Abstract

Since little is known about the detailed situations of falls on Mount Fuji, the aim of this study was to clarify the risk factors of falls on Mount Fuji in Japan. We conducted a questionnaire survey of 556 participants who had climbed Mount Fuji and collected the following information: fall situation, mental status, fatigue feeling, sex, age, climbing experience on Mount Fuji and other mountains, summit success, whether staying at a lodge, use of a tour guide, and symptoms of acute mountain sickness. Among the 556 participants, 167 had a fall (30%). Among 167 participants who had experienced a fall, 30 had fallen more than three times (18%). The main cause (>60%) of fall were slips. The most optimal model using multiple logistic regression (no fall = 0, and fall = 1) found eight significant risk factors, including sex, prior climbing experience on Mount Fuji, staying overnight at a lodge, subjective feeling of relaxation, sleepiness, emotional stability, dullness, and eyestrain. These results suggest that females, people who have no prior climbing experience on Mount Fuji, and people who did not stay at a lodge should pay attention to an increased risk of falls on Mount Fuji.

## 1. Introduction

The incidence of acute mountain sickness (AMS) is known to increase at high altitudes (>2500 m) [1]. Mountain climbing involves level, uphill (ascending), and downhill (descending) walking. In addition to the increased risk of AMS with climbing altitude during the ascent, there are other risks during descents, which should not be ignored. For example, while descending, the center of mass of the whole body is behind the front foot and therefore, outside the base of support, potentially leading to instability, and an increased risk of a slip [2]. It has also been reported that six-week whole-body vibration training significantly improved isometric knee extensors strength capacity (quadriceps muscle) and control of dynamic stability after a slip, resulted in the fall rates compared with no training groups [3]. They suggested that these results may be attributed to the enhanced trunk segment movement control [3]. Additionally, since muscle damage to the lower limbs due to eccentric muscle contractions is common during downhill walking [4], falls may occur more frequently during the descent. Indeed, about 75% of falls during mountain hiking occur during descents [5,6]. Falls could potentially cause more or less serious bruises, sprains, and fractures. These injuries make it difficult for people to move by themselves, and hence, the prevention of falls is important for safe and comfortable climbing. However, comparative studies on falls and the related factors during mountain climbing on various mountains are limited. An epidemiological study has reported that on Denali about 75% of the traumatic injuries requiring rescue could be attributed to falls [5]. Specifically, advancing age, failure to reach the summit, and not being a member of a guided party are related with need for rescue on Denali [5]. Moreover, men and elder age were more likely to have fall-related accidents in the Austrian Alps [6]. In addition, a recent study advocated that better understanding of internal circumstances, such as self-assessment of fatigue, is also important to decrease the risk of falls [6].

Mount Fuji (altitude: 3776 m) is the highest mountain in Japan and is known as a UNESCO World Cultural Heritage site. Therefore, it is the most famous mountain in Japan and some 250,000 people climb it annually. Considering those who have less than three experiences of any mountain climbing as novice climbers, more than 50% of Mount Fuji climbers are novice climbers [7]. Moreover, approximately 60% of climbers on Mount Fuji are first-time climbers [8]. Mountain climbing on uneven and rugged terrain has been shown to pose several challenges, especially for the novice climbers or for those who climb infrequently [9]. A previous study has reported that the most frequent cause of traumatic injuries while descending the Mount Fuji was tripping or slipping [10]. However, the previous study used only the annual reports published by the local Yamanashi Prefectural Police with a relatively small sample size (ca. 150), and analyzed only the prevalence of falls [10]. Therefore, data on several important factors that cause falls were lacking in that study [10].

Accordingly, we hypothesized that climbers with less experience on Mount Fuji and/or who has more subjective fatigue feeling at lower limbs would have more experience to fall. To test this hypothesis, we evaluated several potential factors including sex, age, AMS symptoms, climbing experience, self-assessment of fatigue, single-day or staying overnight at a lodge, summit success, and the use of tour guide that may influence the falls on Mount Fuji.

## 2. Materials and Methods

### 2.1. Survey Site

Mount Fuji has four different climbing routes (Yoshida, Subashiri, Gotenba, and Fujinomiya). We established the questionnaire survey site at the Yoshida trailhead sign (altitude: 2280 m, Figure 1a,b) because more than 60% of all climbers use this route and only a few tourists visit the trailhead for hiking. We did not establish the survey site at the Fuji Subaru Line parking lot (altitude: 2305 m), which could be reached about 20 min by walking from the survey site. The reason was that many tourists (not climbers) visit the parking lot (Figure 1c), and hence, it was difficult to distinguish between tourists and climbers there.

We called out to all people who visited the survey site, and asked whether the persons climbed Mount Fuji. If the persons climbed Mount Fuji, we regarded them (participants) as climbers, not hikers or trekkers, irrespective that they reached the summit. This is because while so many people climb Mount Fuji aforementioned, the ascending route of Mount Fuji has an especially steep section between the 7th (altitude: 2700 m) and the 8th (altitude: 3000 m) steps with rocky place, and the descending route involves a continuous long gravel, sand, and volcanic ash [10]. Moreover, almost climbers aimed to see the sunrise at the top of Mount Fuji [10]. This term, climbers, has also been used in some previous studies that was conducted at Mount Fuji [8,10,11].

### 2.2. Participants

We used data from a questionnaire survey for this study. This study was approved by the Ethics Committee of the Mount Fuji Research Institute in Japan, in accordance with the Declaration of Helsinki. We surveyed the descending climbers who reached the survey site of Mount Fuji on 21–22 July and 15–16 August, 2018. Participants with an age over 18 years old, were surveyed between 08:00 and 12:00. We confirmed that all climbers had reached the parking lot in their own vehicles or a tour bus and that they had begun their descent during the early hours of the morning. Participants who stopped to respond voluntarily and individually were included in the survey, to avoid responses made under any pressure or after consultation with friends. After providing a detailed explanation of the study, informed consent was obtained from each participant. The questionnaire survey sheet was handed over at that point. To avoid several biases, including duplicate responses (climbers who descended a second time on the survey day), as well as vague or rushed responses due to the participant’s busy schedule (such as just before the departure time of the return bus), we confirmed the following criteria for all participants: (1) it was the first response to the survey and (2) there was sufficient time to complete the questionnaire carefully. In our preliminary survey test at our institution, the time to complete all questionnaires was about 10 min. To obtain a sufficient sample size, we used the methodology described in our previous studies [8,11].

### 2.3. Questionnaire

The questionnaire parameters in the present study are shown in Table 1. The participants were asked whether or not they had a fall. In the present study, “fall” was defined as climbers who had a ground contact with any portions of the body (e.g., one hand or one knee or hip). Based on the response they were divided into two groups: (1) had not experienced a fall and (2) had experienced a fall. The questionnaire included questions on the site of the fall (while ascending or descending), causes of the fall (stagger, stumbling, slip, trekking pole, and others), injury status (injured or not), and type of injury (knee pain, sprained ankle, scratch, and others).

The survey also evaluated the following 12 parameters: (1) age, (2) sex, (3) climbing experience on Mount Fuji, (4) climbing experience on any other mountains, (5) the previous night’s sleeping arrangements (single day climbers or overnight stay lodgers), (6) used a tour guide or not, (7) reached the summit or not, (8) with or without AMS, (9) whole body fatigue assessed by the visual analogue scale (VAS), (10) fatigue in the lower limbs assessed by VAS, (11) mood change following acute physical activities, and (12) self-assessment of fatigue. The VAS findings in response to questions 9 and 10 were marked on a straight horizontal line 100 mm in length [the ends were defined as the extreme limits of the parameter] ranging from the best (left, 0) to the worst (right, 100). The question on mood change (# 11) had three subscales: (11-1) subjective- pleasant emotions, (11-2) anxiety, and (11-3) feeling of relaxation. Each of these subscales was composed of four combined sub-questions with seven ordinal scores of -3 to 3 (the scores of each subscale was represented as an average value). Similarly, self-assessment of fatigue (#12) had following five subscales: (12-1) subjective feeling of sleepiness, (12-2) emotional stability, (12-3) uneasiness, (12-4) dullness, and (12-5) eyestrain. Each of these subscales was composed of five combined sub-questions with five ordinal scores of 1 to 5, and the score of each subscale was represented as an average value. Parameters 11 and 12 were surveyed based on previous studies [12,13]. As a result, we obtained 18 parameters that could potentially explain occurrence of a fall. While parameter 1 was a continuous variable, parameters 2–8 were binary selections. Here, “male,” “climbing experience on Mount Fuji is more than once,” “climbing experience on any other mountains is more than two years,” “single day climbers,” “without a tour guide,” “did not reach the summit,” and “without AMS” were all defined as ‘‘0′’. To evaluate the symptoms of AMS, we used the Lake Louise AMS scoring system (LLS) that was newly updated in 2018, and included headache, gastrointestinal symptoms, fatigue and/or weakness, and dizziness/light-headedness [14]. The presence of headache score at least one point and, and a total score of at least three points was diagnosed as AMS [14].

### 2.4. Weather Information

The ambient temperature and relative humidity at the top of Mount Fuji on the survey days were obtained from the data of the Japan Meteorological Agency.

### 2.5. Statistics

To identify the important factors that affect the incidence of falls, we performed multiple logistic regression analysis. The response variable in this analysis was a fall (scored as “1”) or no fall (scored as “0”). Since a high correlation between the explanatory variables can distort the results of multiple logistic regression analysis [15], we checked the multicollinearity among the 18 explanatory variables using variance inflation factors (VIFs), prior to the analysis. The VIF value of an explanatory variable was calculated as “1/(1 − R^2^)” where R^2^ is the usual R-squared from a linear regression explained by the remaining explanatory variables. A high VIF value (>5) for an explanatory variable [16] indicates that the variable was highly correlated with the other variables, and should be omitted. However, no correlation was seen among the explanatory variables (VIF ≤ 3 for all variables), and thus, all variables were used in the analysis. We first built a model, including all the 18 explanatory variables. Then, to find the optimal model containing the parameters that best explain the data, we performed model selection by backward stepwise elimination using Akaike Information Criterion (AIC); where the least important variables were deleted one by one [17]. The AIC evaluates the balance between goodness of fit and model complexity. The AIC value decreases if an explanatory variable that seldom explains the incidence of fall is deleted. Thus, the model with the lowest AIC value can be considered as optimal. The statistical differences between the optimal model and null model were tested using the likelihood ratio test. A *P* value less than 0.05 was defined as statistically significant. Statistical analysis was performed using the free software R version 3.1.3. (Vienna, Austria) [18].

## 3. Results

Based on the data from the Japan Meteorological Agency, the mean ambient temperature throughout 24 h during the four survey days at the summit of Mount Fuji was 8 °C (range: 4–14 °C), and the mean relative humidity was 75% (range: 26–100%) on the survey days. No extreme environmental conditions, such as severe cold, snow, or typhoons were recorded on those days.

In total, 802 Japanese participants aged >18 years were surveyed. Of these, 246 were excluded from further analysis due to lack of information on the survey form. As a result, we obtained 556 valid responses (valid response rate: 69%). The prevalence of “fall” was found in 167 participants (fall rate: 30%), and 30 fell more than three times. Among the participants who had a fall, 50 cases of injuries were reported, which included knee pain, sprained ankle, and scratch. The main cause (>60%) of a fall was a slip.

The attributes of the participants are summarized in Table 2. The number of participants who had “falls” or “no falls” without falls was calculated for each of the categorical variables (sex, prior climbing experience on Mount Fuji, climbing experience on any other mountains, staying at a lodge, accompanying tour guide, summit success and AMS), while for the remaining continuous variables the averages and standard deviations were calculated for each fall category.

The optimal model of the multiple logistic regression included eight explanatory variables: sex, climbing experience on Mount Fuji, staying overnight at a lodge, subjective feeling of relaxation, sleepiness, emotional stability, dullness, and eyestrain (Table 3). Of these, partial regression coefficients other than dullness were significant at the 5% level, but dullness was not significant (*p* = 0.113).

The AIC of the null and the optimal models were 682 and 662, respectively, indicating that the variables included in the optimal model contribute to predicting the occurrence of a fall. The likelihood ratio test revealed that the difference between the two models was statistically significant (null model: d.f. = (0, 555), residual deviance = 680; optimal model: d.f. = (8, 547), residual deviance = 644, *p* < 0.001).

## 4. Discussion

### 4.1. Effect of Sex on Falls

A previous study reported that the sex ratio for non-fatal accidents was similar for males (45%) and females (55%) [6]. This result is inconsistent with our results, possibly, because of different locations and populations. In the present study, the main cause of falls was slips (more than 60%). Since women have lower dynamic postural control compared to men [19], it is still possible that women have more likely to slip. It has been reported that better control of dynamic stability is related to greater quadriceps muscle strength [3]. A previous study demonstrated that women have greater reduction in maximal voluntary contraction at vastus lateralis (one of the components of quadriceps) after step exercise including eccentric exercise, which are similar to work out during mountain descent [20]. These findings may account for the greater fall rate among females in the present study. However, we must acknowledge that this hypothesis is highly speculative as we did not assess the detailed variables such as changes in muscle damage or strength directly. Therefore, future studies to elucidate underlying the mechanisms about the different ratio of fall risks between the sexes are needed.

### 4.2. Effect of Prior Climbing Experience on Mount Fuji on the Incidence of Falls

A previous study concluded that novice climbers should be alerted to the risk of falling [21], and our results seem to support this finding. However, one notable issue is that we did not find any association between prior climbing experience on other mountains and the prevalence of falls on Mount Fuji. These results may indicate that novice climbers on Mount Fuji are likely to have more falls irrespective of their previous experience climbing other mountains. Specifically, Mt. Fuji’s scree have more than 1 m of volcanic ejecta, called scoria with dozens of centimeters in diameter. Furthermore, the descending road is steep (about 30 degrees) and continues for more than 1.5 km. For this reason, many climbers frequently sink their foot sinks into the ground and slips. Under these conditions, a risk of slip, subsequently falls may increase. In Japan, it is rare to descend under such conditions, so it is considered that mountain climbing experience in other mountains is not valid for falls (see Figure 1d) [10]. Moreover, there is a possibility that climbers who had a fall in the past may have chosen not to climb Mount Fuji again, leading to an overestimation of falls among the novice climbers.

### 4.3. Effects of Other Potential Factors on Falls

Participants who did not stay at a mountain lodge were more likely to experience falls. Nédélec et al. showed that sleep deprivation may impair muscle damage repair and lead to an increase in mental fatigue [22]. Therefore, it is possible that they did not recover from muscle fatigue (or damage) ineffectively. In contrast, participants with more sleepiness were not likely to have a fall. It is possible that participants with more sleepiness could not move quickly. However, we must acknowledge this hypothesis is speculative, and future studies should be warranted.

With respect to the effects of other subjective feelings, previous studies have reported that mental fatigue [23] or symptoms of depression [24] are related to a risk for slips and falls, indicating a potential relation between an individual’s mental condition and an increase risk of fall. In the present study, participants with a lower relaxation feeling, higher emotional stability, and more eyestrain were likely to experience a fall. However, it was difficult to clarify the causal relationships between subjective feelings and an increase in the risk of falls. This is because the risk of falls is influenced not only by subjective feelings but also by other confounders such as the mental condition at that moment. More specifically, whether fatigue assessed by VAS, mood change, and subjective fatigue feeling (parameters #9 to #12) can be a predictor or not is questionable because the survey was conducted after climb. Thus, future studies are required. There might be one possible explanation to account for the relationship between higher eyestrain and increase in the risk of fall risks. A moderate correlation has been reported between visual function and sway [25], and sway is strongly associated with the risk of falls [26].

### 4.4. Methodological Considerations

There are several factors that limit the interpretations of our results. First is the small size (*n* = 556). However, in comparison with some previous studies that have used questionnaire surveys at high-altitude on rescues (*n* = 231) [5], 10 fall-related accidents (*n* = 400–700 per year) [6], and AMS (*n* = 130–466) [8,27,28,29,30,31,32,33], our sample size was not small. Second, several potential biases associated with a questionnaire survey study, such as effects of respondent behavior, item characteristics and context, and measurement context, were not considered [34]. For example, the numbers of classification that were used in our optimal model of multiple logistic regression should have been considered. Unlike factors such as sex, prior climbing experience on Mount Fuji, and staying at a lodge, mood change and mental fatigue were classified into seven or five grades. Thus, we are uncertain how a 1-grade difference in each of these variables might have had a significant impact in predicting the risk of falls. Moreover, it should be considered that our survey time was between 08:00 and 12:00 and that we could not assess severity of injury. Thus, we might have missed some of the climbers, especially those who descended earlier (drop out climbers or rapid-descent climbers) or later (climbers with severe fatigue and with many falls) or transported person due to severe injuries. Similarly, as we could not count the numbers of participants who refused to response our survey, we must acknowledge that our results may be affected by these unknown responses. Third, due to the limited numbers of questionnaire parameters, the effect of other potential candidate factors, such as obesity [35], individual maximum gait speed [36], foot wear [37], unexpected road surfaces [38], load carriage [39], use of a trekking pole [9], and risk perception for falls [40], which have been pointed out previously, were not evaluated. Future studies that include these parameters are warranted. Finally, our results were obtained by selected Japanese populations who climbed on Mount Fuji, and thus, whether our results can be generalized into other populations and mountains (e.g., Caucasian in the European Alps.) or not is uncertain.

## 5. Conclusions

We found that sex (female), no prior climbing experience on Mount Fuji, and not staying at a lodge are factors related to an increase in the risk of falls on Mount Fuji. These results are informative for future climbers and would be important for the safety of a large number of climbers on Mount Fuji.

## Figures and Tables

**Figure 1 ijerph-16-04234-f001:**
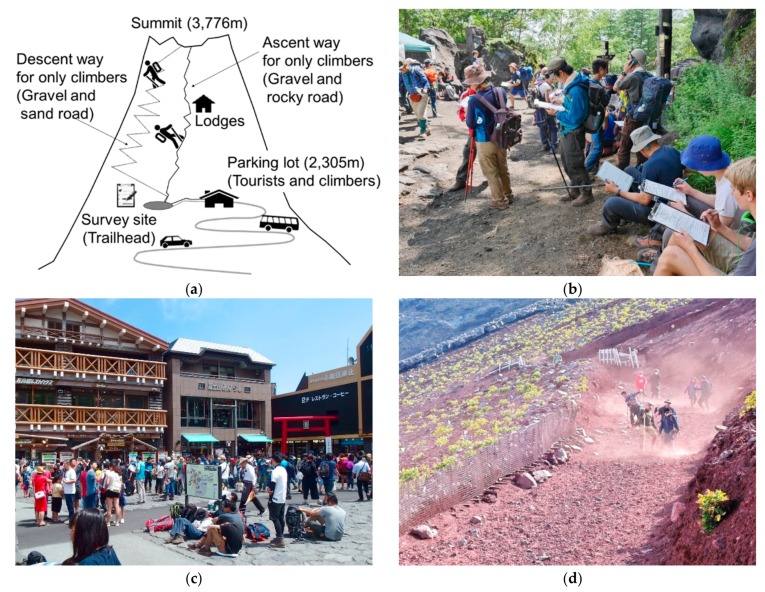
Illustration of the survey locations at Mount Fuji. (**a**) The bold gray line indicates the driving road, solid black line indicates the ascending road, and the dashed black line indicates the descending road. (**b**) A survey scene at the site. (**c**) A square next to the parking lot. (**d**) A fall scenario for climbers while descending on Mount Fuji.

**Table 1 ijerph-16-04234-t001:** Questionnaire used in the present study.

**Please respond to the following questions with respect to falls and their situations**
Did you fall in this mountain climbing?	1. Yes (ground contact with any portions of the body)2. No
First fall situation	Slope	a. Ascent	b. Descent	c. Unknown	d. Other	
Cause	1. Stagger	2. Stumble	3. Slip	4. Caught pole	5. Other
Injury	1. None	2. Knee pain	3. Ankle sprain	4. Scratch	5. Other
Second fall situation	Slope	a. Ascent	b. Descent	c. Unknown	d. Other	
Cause	1. Stagger	2. Stumble	3. Slip	4. Caught pole	5. Other
Injury	1. None	2. Knee pain	3. Ankle sprain	4. Scratch	5. Other
Situation of three or more falls	Times	______ times
Slope and times	a. Ascent______ times	b. Descent______ times			
Cause	1. Stagger	2. Stumble	3. Slip	4. Caught pole	5. Other
Injury	1. None	2. Knee pain	3. Ankle sprain	4. Scratch	5. Other
**Please respond to following questions (parameters of #1 to #7 in the main text).**
Age	______ years old
Sex	1. Male	2. Female
Prior climbing experience on Mount Fuji	1. First time	2. More than once
Climbing experience on other mountains	1. Less than two years	2. More than two years
Staying	1. Overnight stay lodgers	2. Single day climbers
Guide	1. With tour guide	2. Without tour guide
Summit success	1. Reached	2. Failed
**Please tell me about your symptoms of acute mountain sickness a recalling the worst condition. Please mark only one appropriate answer with a circle (parameters #8 in the main text).**
Headache	0. No headache	1. Mild headache
2. Moderate headache	3. Severe incapacitating headache
Gastrointestinal symptoms	0. No gastrointestinal symptoms	1. Poor appetite or nausea
2. Moderate nausea or vomiting	3. Severe incapacitating nausea or vomiting
Fatigue and/or weakness	0. No tired or weakness	1. Mild fatigue/weakness
2.Moderate fatigue/weakness	3. Severe incapacitating fatigue/weakness
Dizziness/lightheadedness	0. No dizzy	1. Mild dizziness
2. Moderate dizziness	3. Severe incapacitating dizziness
**Please indicate how fatigued you feel with a mark ( / ) on the line below (parameters #9 and #10 in the main text; visual analogue scale).**
Whole body	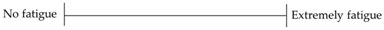
Lower body	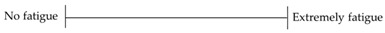
**Please tell me about your current feelings. Please mark only one appropriate answer with a circle (parameter #11 in the main text).**
	Extremely unlikely	Moderately unlikely	Slightly unlikely	Neither	Slightly likely	Moderately likely	Extremely likely
1 I am lively	1	2	3	4	5	6	7
2 I am relaxed	1	2	3	4	5	6	7
3 I feel anxious	1	2	3	4	5	6	7
4 I feel brisk	1	2	3	4	5	6	7
5 I am comfortable	1	2	3	4	5	6	7
6 I am bothered	1	2	3	4	5	6	7
7 I am perky	1	2	3	4	5	6	7
8 I feel calm	1	2	3	4	5	6	7
9 I am regretful	1	2	3	4	5	6	7
10 I feel refreshed	1	2	3	4	5	6	7
11 I feel rested	1	2	3	4	5	6	7
12 I am worried	1	2	3	4	5	6	7
**Please tell me about your current conditions. Please mark only one appropriate answer with a circle (parameter #12 in the main text).**
	Disagree completely	Agree scarcely	Agree slightly	Agree considerably	Agree strongly
1 I feel heavy in the head	1	2	3	4	5
2 I feel nervous	1	2	3	4	5
3 I feel my eyes are dry	1	2	3	4	5
4 I feel ill	1	2	3	4	5
5 I feel restless	1	2	3	4	5
6 I feel a headache	1	2	3	4	5
7 I feel a pain in the eyes	1	2	3	4	5
8 I feel stiff in the neck and shoulders	1	2	3	4	5
9 I feel the brain hot or muddled	1	2	3	4	5
10 I feel like yawning	1	2	3	4	5
11 I feel a pain in the hands or fingers	1	2	3	4	5
12 I feel dizzy	1	2	3	4	5
13 I feel drowsy	1	2	3	4	5
14 I feel a lack of desire to do something	1	2	3	4	5
15 I feel anxious	1	2	3	4	5
16 I feel my eyes are blurry	1	2	3	4	5
17 I feel tired in the whole body	1	2	3	4	5
18 I feel depressed	1	2	3	4	5
19 I feel dullness in the arms	1	2	3	4	5
20 I feel difficulty in thinking	1	2	3	4	5
21 I feel a desire to lie down	1	2	3	4	5
22 I feel eyestrain	1	2	3	4	5
23 I feel a lower back pain	1	2	3	4	5
24 I feel my eyes are blinking	1	2	3	4	5
25 I feel tired in the legs	1	2	3	4	5

The questions are over. We appreciate your cooperation.

**Table 2 ijerph-16-04234-t002:** Characteristics of the surveyed independent variables.

Independent Variables	No Fall	Fall
*Age*	37 ± 14	36 ± 15
*Sex*		
Male	260 (75%)	85 (25%)
Female	129 (61%)	82 (39%)
*Climbing experience on Mount Fuji*		
More than once	142 (77%)	42 (23%)
First	247 (66%)	125 (34%)
*Climbing experience on other mountains*		
More than two years	163 (74%)	57 (26%)
Less than two years	226 (67%)	110 (33%)
*Stay*		
Single day climbers	36 (62%)	22 (38%)
Overnight stay lodgers	353 (71%)	145 (29%)
*Tour guide*		
Without tour guide	249 (71%)	102 (29%)
With tour guide	140 (68%)	65 (32%)
*Summit success*		
Failed	64 (72%)	25 (28%)
Reached	325 (70%)	142 (30%)
*AMS*		
No AMS	298 (71%)	119 (29%)
AMS	91 (65%)	48 (35%)
*Fatigue measured by a visual analogue scale*		
Whole body	67 ± 26	71 ± 26
Lower body	63 ± 27	66 ± 26
*Mood change following acute physical activities* (−3–3)		
Pleasant emotions	1.0 ± 1.4	0.8 ± 1.4
Anxiety	−2.0 ± 1.1	−1.9 ± 1.1
Relaxed	1.0 ± 1.3	0.7 ± 1.3
*Subjective feelings* (1–5)		
Sleepiness	2.7 ± 1.0	2.7 ± 1.0
Emotional stability	1.4 ± 0.6	1.4 ± 0.5
Uneasiness	1.7 ± 0.7	1.7 ± 0.7
Dullness	2.4 ± 0.9	2.6 ± 0.9
Eyestrain	1.8 ± 0.9	1.9 ± 0.9

For the categorical variables, the number of participants belonging to different categories and their proportions (in parentheses) are shown. When analyzing, set the top category to 0 and the bottom to 1. For continuous variables, values are represented as the means ± standard deviations.

**Table 3 ijerph-16-04234-t003:** Summary of the optimal model of multiple logistic regression for “a fall” or “no fall”.

Independent Variables	Partial Regression Coefficient	95% CI		*p* Values
		Lower	Upper	
*Sex*				
Female	0.528	0.134	0.922	0.009
*Climbing experience on Mount Fuji*				
First	0.548	0.122	0.974	0.012
*Staying*				
Overnight stay lodgers	−0.615	−1.215	−0.014	0.045
Subjective feeling of relaxation	−0.212	−0.373	−0.052	0.010
Sleepiness	−0.303	−0.586	−0.021	0.035
Emotional stability	−0.462	−0.883	−0.041	0.032
Dullness	0.240	−0.057	0.536	0.113
Eyestrain	0.313	0.039	0.586	0.025

CI: confidence interval of partial regression coefficient.

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
