# Peer review of "Prevalence of Falls on Mount Fuji and Associated with Risk Factors: A Questionnaire Survey Study"

_ijerph, 2019, doi:10.3390/ijerph16214234_

Round 1

Reviewer 1 Report

The present study investigated the prevalence of falls on Mount Fuji and associated risk factors. However, despite this manuscript presenting clinical relevance for the injury prevention and rehabilitation fields, as well as adequacy of the scope of the Journal, it still needs improvement, which is considered as a prerequisite to a future publication.

Firstly, I think that the introduction of the study can be improved on some points, and that the objectives of the study are not clear. Methods and results need minor adjustments. The discussion needs to be expanded at some points. 

Other considerations are made the following specific evaluation.

TITLE is presented in a clear and concise form, which is consistent to the authors investigation. The authors launch a startling question that draws attention to the reader.

The ABSTRACT includes a brief background and the purpose of the study (to investigate the prevalence of falls and related factors on Mount Fuji). METHODS and RESULTS are well written. The CONCLUSION is directly related to the objectives.

LINE 3 – I suggest modify “following the information” to “following information”.

INTRODUCTION

The authors briefly point out acute mountain sickness (AMS) as a problem in mountain climbing. Authors suggested that greater muscle strength is associated with improving the control of dynamic stability after a slip. I understand that this association is being made very broadly. Please, explain the context on this point and indicate exactly which muscle groups seem most relevant to this statement, establishing cause and effect relationships

The authors problematize the study, but the purpose definition of the work was incomplete. No hypotheses were described. I think the authors could avoid describe methodological elements at the end of the last paragraph. Please review this aspect.

METHODS

In the most part, the methods and procedures for data collection are described clearly and seem appropriate, in sufficient detail to allow others to replicate and build on published results. The methods comply with ethical principles. The authors pointed out the favorable opinion of the Ethics Committee.The analyses seem in most part appropriate to meet the objectives of the study. However, this section is unnecessarily long, especially in the kinematic analysis. Please shorten if possible. Specific considerations are followed.

LINE 80 – I didn't understand why using the term “Since, the questionnaire...”

LINE 84 – Please review the need to use the comma in “Participants with an age over 18 years old, were surveyed between 08 00 and 12 00”.

LINE 94 – Authors could detail the estimated time to complete the questionnaire.

  RESULTS

The main results are pointed out in a concise and precise form, according to the norms of the journal. There is no duplication of information in charts, tables and text. The figures and tables used include the requirements necessary to address the main issues. I suggest that all numerically described results are also described in their percentage frequency. Sometimes the authors do this. In others not. Percentage values ​​give readers a better idea of ​​the distribution of response rates for each variable.

DISCUSSION

The results verified in the present manuscript are at mostly confronted with other previous works. In most part, authors pointed convergences and divergences between the results, and the interpretations of the findings demonstrate ownership in relation to the central theme of the article. Some studies of a similar nature are cited to assist in the foundation of authors’ considerations. However, it is emphasized that the discussion still needs improvements in some key points, as addressed below.

I suggest modify “muscle strengths” to “muscle strength” both in all the main text.

LINES 196-200 – I believe the discussion of the implications of reducing muscle strength for increased risk of falls is very broad. I suggest that the authors review this discussion and improve it.

LIMITATIONS

The authors indicate and/or justify possible limitations of the study, both in relation to the aspect of sample selection, and in relation to the technical and methodological aspects.

CONCLUSIONS

Authors’ conclusions are based on the study results and point to the association between clinical variables studied.

Author Response

RESPONSE TO THE REVIEWER #1

We wish to express our appreciation for your insightful comments, which we believe have significantly helped to improve the presentation of our work. Your comments are written in Italic fonts, and our responses are written in normal fonts, and red fonts. In the main text, we used MS word tracking system as editorial offices recommended. The red fonts with underlines are also consistent with main text. Please note that page and line # are consistent with tracked history with comments (NOT WITHOUT COMMENTS, THAT IS, CLEAN VERSION). We also appreciate that these responses could persuade you.

Reviewer 1:

Comment: The present study investigated the prevalence of falls on Mount Fuji and associated risk factors. However, despite this manuscript presenting clinical relevance for the injury prevention and rehabilitation fields, as well as adequacy of the scope of the Journal, it still needs improvement, which is considered as a prerequisite to a future publication. Firstly, I think that the introduction of the study can be improved on some points, and that the objectives of the study are not clear. Methods and results need minor adjustments. The discussion needs to be expanded at some points.

Other considerations are made the following specific evaluation.

Response: We responded point-by-point to your comments and modified the text. Please see below.

Comment: TITLE is presented in a clear and concise form, which is consistent to the authors investigation. The authors launch a startling question that draws attention to the reader. The ABSTRACT includes a brief background and the purpose of the study (to investigate the prevalence of falls and related factors on Mount Fuji). METHODS and RESULTS are well written. The CONCLUSION is directly related to the objectives.

LINE 3 – I suggest modify “following the information” to “following information”.

Response: We changed “following the information” to “following information” (Page 1 Line 15). Based on your suggestions, we added a brief background in the abstract and changed as follows;

Page 1 Line 12-13

Since little is known about the detailed situations of falls on Mount Fuji, the aim of this study was to clarify the risk factors of falls on Mount Fuji in Japan.

Along with the changed abstract, we deleted some words “assessed by visual analogue scale” as the abstract word is limited within 200 words (Page 1 Line 16).

Comment: INTRODUCTION

The authors briefly point out acute mountain sickness (AMS) as a problem in mountain climbing. Authors suggested that greater muscle strength is associated with improving the control of dynamic stability after a slip. I understand that this association is being made very broadly. Please, explain the context on this point and indicate exactly which muscle groups seem most relevant to this statement, establishing cause and effect relationships.

Response: We added further explanations as follows;

Page 1 Line 35-39

It has also been reported that six-week whole-body vibration training significantly improved isometric knee extensors strength capacity (quadriceps muscle) and control of dynamic stability after a slip, resulted in the fall rates compared with no training groups [3]. They suggested that these results may be attributed to the enhance trunk segment movement control [3].

Comment: The authors problematize the study, but the purpose definition of the work was incomplete. No hypotheses were described. I think the authors could avoid describe methodological elements at the end of the last paragraph. Please review this aspect.

Response: We added the hypothesis and deleted methodological elements from the end of the last paragraph. Please see below and the main text.

Page 2 Line 64-66

Accordingly, we hypothesized that climbers with less experience on Mount Fuji and/or who has more subjective fatigue feeling at lower limbs would have more experience to fall. To test this hypothesis, we evaluated several potential factors including sex, age, AMS symptoms, climbing experience, self-assessment of fatigue, single-day or staying overnight at a lodge, summit success, and the use of tour guide that may influence the falls on Mount Fuji.

We deleted following descriptions.

For this purpose, we used a questionnaire survey because this methodology was appropriate for obtaining numerous responses (Page 2 Line 68-70).

Comment: METHODS

In the most part, the methods and procedures for data collection are described clearly and seem appropriate, in sufficient detail to allow others to replicate and build on published results. The methods comply with ethical principles. The authors pointed out the favorable opinion of the Ethics Committee. The analyses seem in most part appropriate to meet the objectives of the study. However, this section is unnecessarily long, especially in the kinematic analysis. Please shorten if possible. Specific considerations are followed.

Response: Although we might misunderstand the term ‘kinematic analysis’ that you pointed out, after careful discussion with our colleague, we acknowledge that some descriptions are surely lengthy and redundant. Thus, we cut down analysis (statistics) section slightly as below.

Page 6 Line 150- Page 6 Line 163

The response variable in this analysis was a fall (scored as “1”) or no fall (scored as “0”). Since a high correlation between the explanatory variables can distort the results of multiple logistic regression analysis [15], we checked the multicollinearity among the 18 explanatory variables using variance inflation factors (VIFs), prior to the analysis. The VIF value of an explanatory variable was calculated as “1/(1 − R2)” where R2 is the usual R-squared from a linear regression explained by the remaining explanatory variables. A high VIF value (>5) for an explanatory variable [16] indicates that the variable was highly correlated with the other variables, and should be omitted. However, no correlation was seen among the explanatory variables (VIF ≤ 3 for all variables), and thus, all variables were used in the analysis. We first built a model, including all the 18 explanatory variables. Then, to find the optimal model containing the parameters that best explain the data, we performed model selection by backward stepwise elimination using Akaike Information Criterion (AIC); where the least important variables were deleted one by one [17].

Comment: LINE 80 – I didn't understand why using the term “Since, the questionnaire...”

Response: We deleted “Since, the questionnaire requested some personal information,” and simply described as follows;

Page 2 Line 90-91

“This study was approved by the Ethics Committee of the Mount Fuji Research Institute in Japan, in accordance with the Declaration of Helsinki (ECMFRI-02-2014).”

Comment: LINE 84 – Please review the need to use the comma in “Participants with an age over 18 years old, were surveyed between 08 00 and 12 00”.

Response: We added comma between 08 and 00, and between 12 and 00. We also checked entire manuscript (Page3 Line 94, and Page 10 Line 279).

Comment: LINE 94 – Authors could detail the estimated time to complete the questionnaire.

Response: We added following information.

Page 3 Line 104-105

In our preliminary survey test at our institution, the time to complete all questionnaires was about 10 minutes.

Comment: RESULTS

The main results are pointed out in a concise and precise form, according to the norms of the journal. There is no duplication of information in charts, tables and text. The figures and tables used include the requirements necessary to address the main issues. I suggest that all numerically described results are also described in their percentage frequency. Sometimes the authors do this. In others not. Percentage values give readers a better idea of the distribution of response rates for each variable.

Response: Yes, we understood your comments, and thus, we described percentage in addition to counted numbers. Here, the reviewer suggested to add percentage in all variables. However, in some parts, i.e., age, VAS etc. was averaged with an individual raw data. Thus, we could not add percentage in these values in the Table 2. Alternatively, we have described mean values with standard deviations. We hope that our response could be consistent with your comments.

Comment: DISCUSSION

The results verified in the present manuscript are at mostly confronted with other previous works. In most part, authors pointed convergences and divergences between the results, and the interpretations of the findings demonstrate ownership in relation to the central theme of the article. Some studies of a similar nature are cited to assist in the foundation of authors’ considerations. However, it is emphasized that the discussion still needs improvements in some key points, as addressed below.

I suggest modify “muscle strengths” to “muscle strength” both in all the main text.

Response: We changed “muscle strengths” to “muscle strength” throughout the manuscript.

Comment: LINES 196-200 – I believe the discussion of the implications of reducing muscle strength for increased risk of falls is very broad. I suggest that the authors review this discussion and improve it.

Response: We agree the comment. After careful reviewing our first manuscript and previous literatures, at first, we deleted the ref #21. This is because this the results were obtained by eccentric exercise at elbow flexor muscles despite with a large number of samples (~200), but, to avoid confusion, we deleted it. Alternatively, we described previous studies in detail, and sought not to emphasize as the current discussion is only speculation. Please see below and the main text.

Page 9 Line 219- 228

It has been reported that better control of dynamic stability is related to greater quadriceps muscle strength [3]. A previous study demonstrated that women have greater reduction in maximal voluntary contraction at vastus lateralis (one of the components of quadriceps) after step exercise including eccentric exercise, which are similar to work out during mountain descent [20]. These findings may account for the greater fall rate among females in the present study. However, we must acknowledge that this hypothesis is highly speculative as we did not assess the detailed variables such as changes in muscle damage or strength directly. Therefore, future studies to elucidate underlying the mechanisms about the different ratio of fall risks between the sexes are needed.

Comment: LIMITATIONS

The authors indicate and/or justify possible limitations of the study, both in relation to the aspect of sample selection, and in relation to the technical and methodological aspects.

Response: Another reviewer pointed out some lack of information in the method as below.

Comment from the Reviewer #2: “How was a fall defined? For example, persons often slip and have a light ground contact with one hand. Can you give information on the response rate? How many persons refused to fill in the questionnaire? Is there a possible selection bias? It seems that the authors did not determine the severity of the injuries. Can you provide such information? If not, this limitation should be mentioned.

As mentioned above, while the comments from the reviewer #2 are reasonable comments, unfortunately, we could not rule out these biases completely. For example, in the present study, we defined a fall as “have ground contact with any portions of the body” We did not count refused numbers of participants. So, we added some descriptions in the method, table and the discussion (methodological considerations).

Page 3 Line 109-110

Based on the response they were divided into two groups: (1) had no fall and (2) had a fall. In the present study, we defined “fall” as climbers who had a ground contact with any portions of the body (e.g., one hand or one knee or hip).

The statement aforementioned was also included in the Table 1. Please see the table 1 in the main text.

Page 10 Line 278-283

Moreover, it should be considered that our survey time was between 08:00 and 12:00 and that we could not assess severity of injury. Thus, we might have missed some of the climbers, especially those who descended earlier (drop out climbers or rapid-descent climbers) or later (climbers with severe fatigue and with many falls) or transported person due to severe injuries.    Similarly, as we could not count the numbers of participants who refused to response our survey, we must acknowledge that our results may be affected by these unknown responses.

Comment: CONCLUSIONS

Authors’ conclusions are based on the study results and point to the association between clinical variables studied.

Response: We appreciate your comments.

Other responses: We deleted one citation #21 as the ref. may not be suitable into the discussion. Accordingly, the order of references has been changed.

.

Reviewer 2 Report

The authors attempted to identify risk factors for falls in climber of Mount Fuji. Using a questionnaire based survey, they identified eight risk factors for falls.

The present paper is generally well written and structured. Although data are only hardly generalizable to other locations and populations of hikers and climbers, the paper provides interesting information and adds new knowledge. I have a few concerns to address:

How was a fall defined? For example, persons often slip and have a light ground contact with one hand. Can you give information on the response rate? How many persons refused to fill in the questionnaire? Is there a possible selection bias? It seems that the authors did not determine the severity of the injuries. Can you provide such information? If not, this limitation should be mentioned. Interviews were conducted after return from the climb. Especially fatigue and mood may have changed after the fall or were directly affected by the fall. Therefore it is questionable if the can be used as predictors. Please state. A short statement should be added that the results are based on a selected population at this specific location and a generalization to other climbing locations, e.g. in the European Alps, is limited.

Author Response

RESPONSE TO THE REVIWER #2

We wish to express our appreciation for your insightful comments, which we believe have significantly helped to improve the presentation of our work. Your comments are written in Italic fonts, and our responses are written in normal fonts, and red fonts. In the main text, we used MS word tracking system as editorial offices recommended. The red fonts with underlines are also consistent with main text. Please note that page and line # are consistent with tracked history with comments (NOT WITHOUT COMMENTS, THAT IS, CLEAN VERSION). We also appreciate that these responses could persuade you.

Comment: Reviewer 2:

The authors attempted to identify risk factors for falls in climber of Mount Fuji. Using a questionnaire based survey, they identified eight risk factors for falls.

The present paper is generally well written and structured. Although data are only hardly generalizable to other locations and populations of hikers and climbers, the paper provides interesting information and adds new knowledge. I have a few concerns to address:

Response: First of all, we thank that you raised positive comments for our current work.

We sought to respond to all your comments point-by-point as below.

Comment: How was a fall defined? For example, persons often slip and have a light ground contact with one hand. Can you give information on the response rate? How many persons refused to fill in the questionnaire? Is there a possible selection bias? It seems that the authors did not determine the severity of the injuries. Can you provide such information? If not, this limitation should be mentioned.

Response: Yes, regarding to your comments, such as “a fall definition”, “the response rate (otherwise numbers of refused climbers)”, and “the severity of the injuries”, unfortunately, we could not obtain this information, and therefore, we added some descriptions in the method, table, and methodological considerations as below.

Page 3 Line 109-110

Based on the response they were divided into two groups: (1) had no fall and (2) had a fall. In the present study, we defined “fall” as climbers who had a ground contact with any portions of the body (e.g., one hand or one knee or hip).

The statement aforementioned was also included in the Table 1. Please see the main text.

Page 10 Line 278-283

Moreover, it should be considered that our survey time was between 08:00 and 12:00 and that we could not assess severity of injury. Thus, we might have missed some of the climbers, especially those who descended earlier (drop out climbers or rapid-descent climbers) or later (climbers with severe fatigue and with many falls) or transported person due to severe injuries.   Similarly, as we could not count the numbers of participants who refused to response our survey, we must acknowledge that our results may be affected by these unknown responses.

Comment: Interviews were conducted after return from the climb. Especially fatigue and mood may have changed after the fall or were directly affected by the fall. Therefore it is questionable if the can be used as predictors. Please state.

Response: We already wrote as below, but, based on your suggestions, we further described. Please see below and the main text.

Page 9 Line 257-261 (in the revised manuscript)

“In the present study, participants with a lower relaxation feeling, higher emotional stability, and more eyestrain were likely to experience a fall. However, it was difficult to clarify the causal relationships between subjective feelings and an increase in the risk of falls. This is because the risk of falls is influenced not only by subjective feelings but also by other confounders such as the mental condition at that moment.”

In addition to these statements above mentioned, we added further limitation followed by “as the mental condition at that moment”.

Page 9 Line 261-263

More specifically, whether fatigue assessed by VAS, mood change, and subjective fatigue feeling (parameters #9 to #12) can be a predictor or not is questionable because the survey was conducted after climb. Thus, future studies are required.

Comment: A short statement should be added that the results are based on a selected population at this specific location and a generalization to other climbing locations, e.g. in the European Alps, is limited.

Response: We added a brief description in the last paragraph of methodological considerations as follows;

Page 10 Line 291-293

Finally, our results were obtained by selected Japanese populations who climbed on Mount Fuji, and thus, whether our results can be generalized into other populations and mountains (e.g., Caucasian in the European Alps.) or not is uncertain.  

Other responses: We deleted one citation #21 as the ref. may not be suitable into the discussion. Accordingly, the order of references has been changed.

Reviewer 3 Report

Congratulate the authors for the study done

Author Response

RESPONSE TO THE REVIWER #3

We really appreciated that you took time for reviewing with your efforts.

Comment: Reviewer 3; Congratulate the authors for the study done.
